



# Distribution and seasonal evolution of supraglacial lakes on Shackleton Ice Shelf, East Antarctica

Jenifer F. Arthur [1], Chris R. Stokes [1], Stewart S.R. Jamieson [1], J. Rachel Carr [2], Amber A. Leeson [3]

[1]Department of Geography, Durham University, Durham, DH1 3LE, UK
[2]School of Geography, Politics and Sociology, Newcastle University, Newcastle-upon-Tyne, NE1 7RU, UK
[3]Lancaster Environment Centre/Data Science Institute, Lancaster University, Bailrigg, Lancaster, LA1 4YW, UK

*Correspondence to*: Jennifer F. Arthur (jennifer.arthur@durham.ac.uk)

**Abstract.** Supraglacial lakes (SGLs) enhance surface melting and can flex and fracture ice shelves when they grow and
subsequently drain, potentially leading to ice shelf disintegration. However, the seasonal evolution of SGLs and their influence
on ice shelf stability in East Antarctica remains poorly understood, despite some potentially vulnerable ice shelves having high
densities of SGLs. Using optical satellite imagery, air temperature data from climate reanalysis products and surface melt
predicted by a regional climate model, we present the first long-term record (2000-2020) of seasonal SGL evolution on
Shackleton Ice Shelf, which is Antarctica's northernmost remaining ice shelf and buttresses Denman Glacier, a major outlet
of the East Antarctic Ice Sheet. In a typical melt season, we find hundreds of SGLs with a mean area of 0.02 km$^2$, a mean depth
of 0.96 m, and a mean total meltwater volume of 7.45 x10$^6$ m$^3$. At their most extensive, SGLs cover a cumulative area of 50.7
km$^2$ and are clustered near to the grounding line, where densities approach 0.27 km$^2$ per km$^2$. Here, SGL development is linked
to an albedo-lowering feedback associated with katabatic winds, together with the presence of blue ice and exposed rock.
Although below average seasonal (December-January-February, DJF) temperatures are associated with below average peaks
in total SGL area and volume, warmer seasonal temperatures do not necessarily result in higher SGL areas and volumes.
Rather, peaks in total SGL area and volume show a much closer correspondence with short-lived high magnitude snowmelt
events. We therefore suggest seasonal lake evolution on this ice shelf is instead more sensitive to snowmelt intensity associated
with katabatic wind-driven melting. Our analysis provides important constraints on the boundary conditions of supraglacial
hydrology models and numerical simulations of ice shelf stability.

## 1 Introduction

Supraglacial lakes (SGLs) form when snow and ice melt ponds on glaciers and ice sheets (Echelmeyer et al., 1991). SGLs are
widespread around the margins of Antarctica and can influence ice shelf dynamics both directly and indirectly (Arthur et al.,
2020; Kingslake et al., 2017; Stokes, et al., 2019). Specifically, ice shelves flex when SGLs pond and drain on their surfaces,
which can weaken them and make them more prone to break-up (Banwell, et al., 2013; Banwell et al., 2019; Banwell and
MacAyeal 2015). An often cited example of this was the appearance and abrupt drainage of >3,000 SGLs on the surface of
the Larsen B Ice Shelf in 2002 in the days preceding its collapse (Glasser and Scambos 2008; Leeson et al., 2020), triggering





a chain reaction of SGL hydrofracture and drainage events (Banwell et al., 2013; Robel and Banwell 2019). Thus, the evolution and interaction of SGLs on ice shelves has important consequences for the stability of ice shelves, many of which exert a buttressing effect on inland ice flow (Fürst et al., 2016).

Recent research has shown SGLs form part of active surface meltwater systems on the surface of numerous Antarctic outlet glaciers and ice shelves (Banwell et al., 2019; Bell et al., 2017; Langley et al., 2016; Lenaerts et al., 2017; Moussavi et al., 2020; Kingslake et al., 2017; Stokes et al., 2019) and are particularly widespread on the Antarctic Peninsula and around the margins of the East Antarctic Ice Sheet (Arthur et al., 2020; Kingslake et al., 2017; Stokes et al., 2019). However, few studies

have conducted detailed analyses of the seasonal behaviour and spatial distribution of SGLs on ice shelves in East Antarctica. As such, direct observations of lake interactions and drainage are currently constrained to a limited number of ice shelves (Banwell et al., 2019; Bell et al., 2017; Langley et al., 2016; Liang et al., 2019; Moussavi et al., 2020; Kingslake et al., 2015), and our understanding of the importance of SGL evolution on ice-shelf-instability remains limited.

Here, we focus on Shackleton Ice Shelf, which is East Antarctica's sixth largest and most northerly remaining ice shelf in Antarctica (Rignot et al., 2013; Zheng and Zhou, 2019). Over 70% of the ice shelf area exerts a buttressing effect on the major outlet glaciers flowing into it and a number of ice rises and islands act as pinning points for the ice shelf (Fürst et al., 2015; 2016; Fig. 1). The remaining 30% of the ice shelf has been identified as passive shelf ice, i.e. that could be removed without acceleration of inland ice (Fürst et al., 2016). The ice shelf is fed by Denman Glacier, whose catchment holds ice equivalent

to ~1.5 m of sea-level and is ~1.9 times larger than the catchment of Pine Island Glacier (Morlighem et al., 2019; Stephenson and Zwally, 1988; Willis et al., 2016). Portions of the ice shelf experience high mean surface melt rates (>200 mm w.e. yr$^{-1}$) compared to other parts of coastal East Antarctica, such as the Fimbul (> 80 mm w.e. yr$^{-1}$), Riiser-Larsen (> 70 mm w.e. yr$^{-1}$), Nivlisen (> 80 mm w.e. yr$^{-1}$), and Roi Baudouin (> 150 mm w.e. yr$^{-1}$) ice shelves (Trusel et al., 2013). High densities of SGLs in the ice shelf grounding zone suggest that it contains ice-saturated firn (Alley et al., 2018; Lenaerts et al., 2017). The

combination of its topographic setting, intense surface melting and low firn air content (FAC) make Shackleton Ice Shelf potentially vulnerable to hydrofracturing (Alley et al., 2018; Stokes et al., 2019).

Our overall aim is to investigate the long-term distribution and evolution of SGLs on Shackleton Ice Shelf. Our first objective is to analyse the intra-seasonal and inter-seasonal evolution in the spatial distribution, extent and volume of SGLs over multiple

melt seasons from 2000 to 2020 using remote sensing. Our second objective is to explore potential glaciological and climatic controls on SGL formation and distribution on this ice shelf in order to improve our understanding of its potential vulnerability.





## 2 Study site

Shackleton Ice Shelf, in Queen Mary Land, East Antarctica (65 ºS; 100 ºE) covers an area of 33,820 km$^2$ (Zheng & Zhou, 2019, Fig. 1). Denman Glacier flows into the ice shelf at speeds over 1800 m a$^{-1}$ at its grounding line (Rignot et al. 2019) and
occupies a trough deeper than 3500 m b.s.l which connects to the Aurora subglacial basin (Morlighem et al., 2019; Roberts et al., 2011). Denman Glacier has accelerated 16% since the 1970s (Rignot et al., 2019) and is thinning inland (Flament and Rémy 2012) and undergoing grounding line retreat (Konrad et al., 2018). The ice shelf itself has accelerated by 43% between 1957 and 2016 (Rignot et al., 2019). Several smaller outlet glaciers drain into the ice shelf, including the Scott, Apfel, Remenchus and Northcliffe Glaciers (Fig. 1). The Bunger Hills (a large nunatak) and an extensive blue ice area lie adjacent to
the ice shelf grounding zone (Fig. 1). Ice thickness varies across the ice shelf from 231 m (at the calving front) to < 10 m around Mill Island (Fretwell et al., 2013; Wearing et al., 2015). Thinning of Shackleton Ice Shelf at a rate of 0.12 m yr$^{-1}$ between 2003 and 2008 has been attributed to enhanced basal melting (Pritchard et al., 2012). SGLs on Shackleton Ice Shelf have been previously mapped for the year 2015 from Landsat 8 imagery, to assist interpretation of snowmelt detection from microwave radiometer and scatterometer data (Zheng and Zhou 2019), but no studies of seasonal and/or interannual SGL
evolution currently exist for the ice shelf.

## 3 Data and methods

### 3.1 Satellite imagery acquisition and processing

For the years between 1974 and 2020 we acquired all Landsat 1, 4 and 5 MSS (Multispectral Scanner), TM (Thematic Mapper),
Landsat 7 ETM+ (Enhanced Thematic Mapper), Landsat 8 OLI (Operational Land Imager) and Sentinel 2A/B MSI (Multispectral Instrument) imagery over Shackleton Ice Shelf during the austral summer months (November to February) with ≤ 20% cloud cover from the USGS EarthExplorer (https://earthexplorer.usgs.gov) and the European Space Agency Copernicus Open Access Hub (https://scihub.copernicus.eu/dhus; Supplementary Table 1). For scenes that included cloud cover within the ≤ 20% threshold, it was mostly restricted to the outermost parts of the ice shelf or further inland beyond the grounding line,
where cloud free scenes indicated that no lakes form. Eight scenes with cloud cover higher than 20% were used, where cloud cover did not obscure most of the ice shelf and SGLs were visible. These were used to analyse the spatial distribution of lakes across the ice shelf but not in our analysis of lake area and volume (Sect. 3.3). The multispectral bands of the Landsat-8 images were pan-sharpened to 15 m using the panchromatic Band 8 and a Brovey Transform (Gillespie et al., 1987) in QGIS. For more recent years (2017 to 2019), Sentinel 2A/B imagery was used in preference to Landsat 8 or another sensor, such as the
Moderate Resolution Imaging Spectroradiometer (MODIS) or Advanced Spacebourne Thermal Emission and Reflectance Radiometer (ASTER), owing to its high spatial resolution (10 m) and revisit period of 5 days.



## 3.2 Automated mapping of SGL extent

We used the Normalised Difference Water Index adapted for ice (NDWI; Yang and Smith 2013) to classify pixels as 'water' or 'non-water' (i.e. ice, snow, exposed bedrock or ocean) following Eq. (1):

$$NDWI_{ice} = \frac{(Blue - Red)}{(Blue + Red)} \qquad\qquad (1)$$

This version of the NDWI has been widely applied to mapping surface meltwater in Antarctica (Arthur et al., 2020; Banwell et al., 2019; Bell et al., 2017) and Greenland (Doyle et al., 2013; Williamson et al., 2017; Yang and Smith 2013). We found NDWI$_{ice}$ produced fewer false SGL classifications over blue ice/slush areas than with other versions of the NDWI, such as those using the near-infrared band (Supplementary Fig. 1).


The first step in our procedure for mapping SGLs was to apply a bedrock mask derived from Landsat 8 data (Burton-Johnson et al., 2016). Exposed bedrock appears spectrally similar to water in the red band, so masking bedrock avoids mis-classification of rock as water. This was particularly important in our study given the large area of nunataks (Bunger Hills) adjacent to the ice shelf. After rock masking was performed, we applied an NDWI threshold value of 0.25, following previous studies

(Banwell et al., 2019; Bell et al., 2017; Dell et al., in review; Doyle et al., 2013; Williamson et al., 2017; Yang and Smith 2013), meaning pixels with NDWI$_{ice}$ > 0.25 were assumed to be water-covered. This threshold was found to be the most appropriate for detecting lakes, apart from eleven scenes, where the threshold value was manually tuned in 0.01 increments to improve SGL identification by including pixels which were visibly water-covered in true-colour composites of the area (Supplementary Table 1).


Once an NDWI threshold value was applied, each image was reclassified into a binary image and vectorised (converted to polygon shapefiles). Clouds were masked by manually by inspecting each RGB image and deleting any cloud. Some manual editing of SGL extents was necessary where floating ice in the centre of SGLs had resulted in pixels being mis-classified as water. The transition from melting ice to saturated firn, slush, shallow ponds and to deeper SGLs makes it challenging to adopt

a clear binary definition of 'lake' versus 'non-lake' (Dell et al., in review). To account for this uncertainty we used a minimum size threshold of two pixels, in order to remove very small SGLs likely comprised solely of mixed pixels (i.e. 200 m$^2$ for Sentinel, 450 m$^2$ for Landsat 7/8, 1800 m$^2$ for Landsat 4/5, 7200 m$^2$ for Landsat 1), following previous studies (Moussavi et al., 2020; Pope et al., 2016, Stokes et al., 2019). For the purposes of our study, we chose not to classify SGLs separately to supraglacial meltwater channels, given the ambiguity in distinguishing between channels and thin, elongate SGLs.


Finally, NDWI classifications were checked against RGB composites and false positives, including small bedrock outcrops not included in the bedrock mask. Dark crevasses were also manually removed. To further reduce noise and improve the visual clarity of the dataset, we then applied the dissolve function in ArcMap to clean up overlapping pixels and the aggregate tool



to combine pixels within a distance of two pixels. Apart from potentially reducing large numbers of very small SGLs, these
changes were largely cosmetic and are insignificant in terms of the uncertainties associated with identifying SGLs and
calculating total SGL area, described in Sect. 3.4.

### 3.3 Extracting lake areas and depths

It is important to remove the influence of different satellite imagery acquisition conditions when extracting SGL extents and
depths. To do this, Digital Numbers were converted to Top of Atmosphere (TOA) reflectance for red and blue bands using the
DOS1 (Dark Object Subtraction) atmospheric correction (Moran et al., 1992) in QGIS and the scene-specific metadata file
containing reflectance multiplicative and additive scaling factors and solar elevation angle. This process removes the cosine
effect at different solar zenith angles (due to different scene time acquisitions) and compensates for differences in solar
irradiance.

We applied a physically-based radiative transfer model to calculate the water depth of all pixels classified as lake (Pope et al.,
2016; Sneed and Hamilton 2007). This method calculates lake water depth (z) using the rate of light attenuation in water, lake-
bottom albedo, and optically-deep water reflectance (Philpot, 1989) following Eq. (2):

$$z = \frac{[\ln(A_d - R_\infty) - \ln(R_z - R_\infty)]}{g} \tag{2}$$

where $A_d$ is lake bed reflectance, $R_\infty$ is the reflectance of optically deep water, $R_z$ is the red band reflectance value of a water-
coloured pixel and g is the attenuation co-efficient rate.

The radiative transfer model was applied to a subset area of the ice shelf (48 x 48 $km^2$, Fig. 1) to allow lake depth comparisons
between years with different satellite imagery coverage. This subset area was selected as the region in which most SGLs were
concentrated during each melt season with consistent coverage of cloud-free imagery, enabling us to extract changes in lake
area and depth on both the floating ice shelf and the grounded ice upstream of its grounding line. Lake area and depth statistics
presented in Sect. 4 are all extracted from this subset area. We excluded nine image scenes from our analysis which did not
fully cover this subset area, to maintain full comparability between individual dates. Issues with cloud cover resulted in only
one useable scene during some melt seasons, and no suitable imagery in others (2002, 2004-2006; Supplementary Table 1).

$A_d$ was calculated image-by-image by extracting the mean red band reflectance of a two-pixel buffer around each lake polygon,
following Banwell et al., (2019). Red light attenuates more strongly in water than blue light, meaning that there are larger
measurable changes in red reflectance over water than for other wavelengths (Box and Ski 2007; Pope et al., 2016). The
reflectance value immediately beyond identified lake areas is assumed to be comparable to the lake bed if it were exposed to
the atmosphere (Pope et al., 2016). $R_\infty$ is typically approximated as pixel reflectance from open ocean water. Most scenes used



in this study did not contain optically-deep water, and we found the average difference between using an $R_\infty$ of zero and using $R_\infty$ values averaged from fifteen nearby scenes to be <10% (Supplementary Information). Therefore, we used an $R_\infty$ of zero for all scenes, following MacDonald et al. (2018) and Banwell et al. (2014; 2019).

For Sentinel 2 scenes, we used Band 4 reflectances for $R_z$ and a g value of 0.8304, following Williamson et al. (2017). For

Landsat 8 scenes, we followed the recommendation of Pope et al. (2016) and took an average of the depths calculated using the red (Band 4) and panchromatic (Band 8) band TOA reflectances, using g values of 0.7507 and 0.3817 respectively. For Landsat 7 ETM+ scenes, we used Band 3 reflectances for $R_z$ and a g value of 0.8049 following Pope et al. (2016). We are unable to calculate lake depths before the year 2000 (i.e. from Landsat 1 MSS, Landsat 4 TM and Landsat 5 TM scenes) using this physically-based depth method because lab-based absorption coefficients within the full red band wavelength ranges (Pope

and Fry 1997) are unavailable. Lake depths within the subset area were extracted using zonal statistics in ArcMap. Lake volumes were calculated by multiplying each pixel area by its calculated water depth within the lake boundaries.

### 3.4 Errors and uncertainties in extracting lake areas and depths

We quantified the uncertainties in our mapping technique by quantitatively comparing SGL areas derived from NDWI with those derived from manual digitisation in four sample areas (Supplementary Fig. 2a-d). The smaller SGLs (<0.01 km$^2$)

generated the largest percentage differences, and we found a mean absolute error of 0.007 and a root mean square error of 0.029 between manually-digitised SGL areas and those derived from NDWI. We attribute this to manual digitisation being less conservative at 'diffuse', less well-defined SGL edges. Despite these differences, we found very close agreement between the two methods ($R^2 = 0.979$, p = 0.006). We also quantified the impact of sensor resolution on lake detection and lake extents, and found there is generally a very good agreement between lake areas derived from Landsat 8 and Sentinel 2 (Supplementary

Fig. 3). We therefore assign a conservative uncertainty of 1% to our total SGL area for each time step, following Stokes et al. (2019).

The depth-reflectance algorithm used to extract lake depths assumes that lake-bottom albedos are homogenous, that lake water contains minimal dissolved matter and that there is no scattering of light from the lake surface associated with wind-driven

roughness (Sneed and Hamilton 2007). We acknowledge that the last of these assumptions may not always hold if there are wind-driven surface waves. Where lakes were partially ice-covered, we could not calculate the depth of the central, probably deepest, regions of these lakes. In such cases, we calculated the depth of the outer ring of lake-ice free water, following Banwell et al. (2014). This is likely to underestimate the volumes of deeper lakes, but we record maximum lake volumes that are similar (i.e. not substantially lower) to those reported elsewhere in Antarctica (Bell et al., 2017; Banwell et al., 2014; Langley et al.,

185    2016).



### 3.5 Extraction of lake elevations, slopes, densities and ice surface velocities

Lakes were classified as forming on grounded or floating ice using the Making Earth System Data Records for Use in Research
Environments (MEaSUREs) grounding line which is derived from an average of differential interferometric synthetic-aperture
radar (inSAR) data from 1992 to 2014 (Rignot et al., 2016). Although a more recent inSAR grounding line exists for parts of
Denman and Apfel Glaciers, derived from ERS-1/2 and Sentinel 1 data acquired in 1996, 2015 and 2017 (DLR, 2018), this is
incomplete on Shackleton Ice Shelf, and the MEaSUREs grounding line was therefore used as the most recent continental-
wide grounding line. Surface elevation and slope values were extracted from the Reference Elevation Model of Antarctica
(REMA), which has an 8 m spatial resolution. Ice velocity values were extracted from the MEaSUREs InSAR-based ice
velocity mosaic which was derived by averaging data covering the period 1996-2016 (Rignot et al., 2017). Lake area densities
were calculated using lake polygon centroids as inputs to the Point Density tool in Arcmap. This tool calculates the cumulative
area of SGLs within 1 $km^2$ cells by dividing the total number of points that fall within a defined neighbourhood around each
raster cell by the area of the neighbourhood. Stokes et al. (2019) apply this method across the East Antarctic Ice Sheet using a
50 km search radius. Here, a 20 km search radius was chosen as an appropriate size given the area of the ice shelf.

### 3.6 Extracting near-surface temperature, surface snowmelt and firn air content

To compare lake areas with surface climatological conditions, we used near-surface air temperature and modelled surface
snowmelt. In the absence of any automatic weather stations within 80 km of Shackleton Ice Shelf, we used monthly averaged
near-surface temperature (2 m temperature) from the European Centre for Medium-Range Weather Forecasts ERA5 reanalysis
from 1979 to 2019 as the best available option. We extracted near-surface temperature over a ~262 x 370 km box covering the
whole ice shelf (Fig. 1). ERA5 reanalysis is provided at 0.25 degree (~31 km) resolution. To investigate whether timings of
modelled snowmelt events and extensive SGL ponding were comparable, we extracted mean snowmelt over the ice shelf from
daily surface melt flux outputs for the period 2000 to 2019 generated by the Regional Atmospheric Climate Model (RACMO)
version 2.4p1, which has a horizontal resolution of 27 km. Details of the model can be found in Van Wessem et al. (2018). To
investigate the relationship between lake areas and simulated firn air content (FAC), we extracted mean and minimum FAC
(total integrated pore space, i.e. the thickness of the equivalent air column contained in the firn in metres) over the ice shelf
from daily FAC for the period 2000 to 2019 generated by RACMO version 2.4p1.

### 3.7 Hydrological routing analysis

To estimate potential meltwater routing across the ice shelf surface (Leeson et al., 2012; Bell et al., 2017) and compare this to
mapped distributions of SGLs, a D-infinity algorithm (Tarboton, 1997) was applied to a 100-metre resolution version of the
REMA DEM (clipped to Shackleton Ice Shelf). We calculated flow direction and accumulated flow into each cell after filling





closed basins (sinks) in the DEM. The output raster was then vectorised to show the potential pathways of surface meltwater routing across the ice shelf once surface depressions have been filled.

## 4 Results

### 4. Spatial distribution of SGLs

We find that SGLs tend to cluster around the ice shelf grounding zone, near exposed blue ice areas and the Bunger Hills nunataks, whereas they are almost entirely absent towards the ice shelf calving front, where surface melt rates are highest (Trusel et al., 2015; Fig. 2a). Lakes form in particularly high densities on Scott, Apfel and Remenchus Glaciers. Indeed, lake area densities reach their maximum (0.27 km$^2$ per km$^2$) on this portion of the ice shelf (Fig. 2b). No lakes form on the heavily crevassed tongue of Denman Glacier. On the ice shelf itself, most lakes are only recorded in a given position on a single date in a particular year, because they advected downstream with ice flow (Fig. 2b-d). The largest recorded feature is an area of coalesced lakes covering up to 11.96 km$^2$ which formed in January 2020 on Scott Glacier, alongside more linear, elongate lakes which occupy a ~10 m deep longitudinal depression on the glacier surface (Fig. 2d). Further west, smaller lakes occupy crevasses in the suture zone between Denman and Scott Glaciers (Fig. 2c). Though the spatial distribution of lakes on these parts of the ice shelf remains similar during each melt season, lakes gradually become advected downstream with ice flow each year (Fig. 2f).

Upstream of the grounding line, lakes have a high tendency to reoccur annually in the same locations (Fig. 2d). Specifically, seven lakes upstream of Apfel Glacier were present in 62% of scenes mapped (Fig. 2d). Just upstream of this cluster of lakes represents the upper elevation limit (527 m a.s.l) of recorded SGLs, 12.8 km inland from the grounding line (~90 km from the coastline). Lakes only formed at this maximum elevation during the six melt years when total lake volume was highest (2010, 2014, 2015, 2017, 2018 and 2020; see Sect. 4.2).

The concentration of SGLs around the ice shelf grounding zone is reflected in their topographic distribution, which is strongly skewed towards low (≤10 m) elevations and low (<1°) ice surface slopes close to and just beyond the grounding line (Fig. 3a-b). SGLs preferentially form on slower-moving ice (<50 m yr$^{-1}$), typically occupying smaller tributary glaciers and suture zones between glaciers rather than on fastest-flowing portions of the ice shelf like Denman Glacier (Fig. 3c). Lakes form in very close proximity to exposed bedrock (<10 m) and are within or downstream of blue ice areas around the ice shelf grounding line (Fig. 3d).



## 4.2 Seasonal evolution of SGLs

### 4.2.1 Intra-seasonal and inter-seasonal lake evolution

To investigate the seasonal behaviour of SGL formation and evolution on Shackleton Ice Shelf, we present time series of total lake area, depth and volume over the 2000-2020 melt seasons (Fig. 4). In a typical melt season, we observed hundreds of SGLs

that were, on average, 0.02 km$^2$ in area, 0.96 m deep, and held a total meltwater volume of 7.45 x10$^6$ m$^3$. The earliest that meltwater starts being stored on the ice shelf surface within SGLs is in late November (Fig. 4, Table 1). This is earlier than has been reported in other regions of East Antarctica, where SGLs have been observed forming from mid-December, but typically begin forming in early January (Langley et al., 2016; Leppäranta et al., 2013; Moussavi et al., 2020).

Total lake area on the ice shelf increases through December/January and peaks in early- to late-January (Fig. 4a), and then decreases through the remainder of the melt season until late February. Mean and maximum lake depths follow a similar seasonal pattern to lake area, and the deepest lakes are generally recorded between early- to mid-January (Fig. 4b). However, mean and maximum lake depths do not always peak simultaneously with lake area (Fig. 4b, Table 1). For example, in 2020, lake depth peaks 30 days before total lake area and volume peak on 1st January. SGLs on grounded ice upstream of the ice

shelf have a higher mean depth of 0.94 m (st dev: 0.68, p = 0.01) compared to a mean depth of 0.91 m (st dev: 0.54) on the ice shelf (Supplementary Fig. 4). Total meltwater volumes remain low (<5 x10$^6$ m$^3$) during the months of November, December and February and peaks between early- to late-January (Fig. 4c). In 2014, 2015 and 2020, lake volume was more than two standard deviations from the 2000-2020 mean. In 2020, total lake area also deviated by almost three standard deviations from the 2000-2020 mean (Fig. 5b). Total lake area was up to 3.9 times higher and total lake volume up to 4.5 times higher than the

long-term mean in the five years 2014, 2015, 2017, 2018 and 2020. At the time of peak total lake volume during these years, lakes covered 0.05%, 0.06%, 0.03%, 0.04% and 0.09% of the total ice shelf area, respectively.

We document lakes forming during three years prior to 2000 where suitable imagery was available (1974, 1989 and 1991), though we were unable to include these in our depth and volume analyses (see Sect. 3.3). During these years, lakes occupy

many of the same locations as in later melt seasons, including on Scott and Apfel glaciers, and a smaller blue ice area near the northern part of the ice shelf grounding line. SGLs are less extensive during these years than in 2000-2020 (total area 20.3 km$^2$, 5.6 km$^2$ and 8.8 km$^2$ respectively), though we note imagery was limited to the very start (early November) and end (late February) of the melt season.

### 4.2.2 Lake drainage events

Lakes disappear in three ways: supraglacial drainage via channels over the ice surface, drainage into the firn, or refreezing and/or snow burial (Fig. 6). Lakes commonly refreeze at the end of a melt season, as indicated by relict frozen lake scars based on their similarity to refrozen lakes in previous studies (e.g. Langley et al., 2016; Leeson et al., 2020; Tuckett et al., 2019; Fig.



6a-c). This is the dominant mechanism of lake disappearance and is commonly preceded by ice 'lids' that grow outwards from lake centres (Fig. 6a-c).


Earlier in the melt season, some lakes drained supraglacially into others via surface channels (Fig. 6d-f). During the peak of the melt season in some years, portions of the ice shelf become saturated and pond into large lakes, such as on the tongue of Scott Glacier where it flows into the ice shelf. This occurred when total lake area exceeded 10 km$^2$ (in February 2004, January 2014, 2015, 2017, 2018 and 2020). During these seasons of more extensive surface melt, supraglacial rivers develop by mid-

January, fed by lakes upstream of the grounding line (e.g. Fig. 6g). The largest of these rivers reaches ~14 km on Apfel Glacier, though these feed other lakes rather than exporting meltwater off the ice shelf edge, as has been observed on Nansen Ice Shelf in Antarctica (Bell et al., 2017).

We also recorded several instances of lakes apparently draining englacially over a ≤ 7-day period during January 2018 and

2020 (Fig. 6h-l). For example, one chain of lakes (Fig. 9k-l) lost ~36% of their 2.2 km$^2$ total area and ~24% of their 3.5 x10$^6$ m$^3$ total volume in ≤ 7 days. This chain of lakes was surrounded by lakes that did not change in area over this period, suggesting they did not refreeze. Neither did downstream lakes grow in size, and nor was this chain of lakes associated with any major streams/rivers which could have exported the meltwater supraglacially. In other years, this same chain of lakes remains undrained and freezes over (as in Fig. 6a-c).


### 4.2.3 Lake evolution and relationship with climate

We compare total lake area during the 2000-2020 melt seasons with mean near-surface temperature extracted from ERA5 reanalyses to investigate the relationship between seasonal lake coverage and temporal fluctuations in air temperature on

Shackleton Ice Shelf (Fig. 7). We find that warmer melt seasons do not necessarily correlate with more extensive lake coverage. For example, although 2019/2020 records the highest total lake area (28.55 km$^2$), it has a lower mean December-January-February (DJF) near-surface temperature (-9.3 °C; Fig. 7c). Similarly, in January 2015 and 2014, the next most extensive lake years, mean DJF temperature was also low (-8.4 °C and -8.6 °C respectively), although lakes started to form earliest in the melt season (19th November 2015 and 2nd December 2014; Fig. 7b). However, in the melt seasons with the five lowest mean DJF

temperatures (1999-2000, 2007-2008, 2011-2012, 2012-2013 and 2015-2016), total lake area did not exceed 5 km$^2$ (Table 2, Fig. 7b). Likewise, we recorded a relatively high maximum lake extent in 2016-17 (10.29 km$^2$), which was the melt season with the highest mean DJF temperature (-6.4 °C; Fig. 7c). Overall, from 2000-2020, total lake area is only weakly positively correlated with mean surface air temperature in the preceding part of the melt season, i.e. from 1st November to the date of lake observation ($r^2$= 0.15, p= 0.04; Fig. 8). Therefore, peak years in lake coverage exhibit only a weak relationship with near-

surface temperature, suggesting that other factors such as localised albedo interactions may act acts as more important controls.





To investigate whether there was a correspondence between peak lake area and volume with surface melt rates (which include meltwater available to refreeze in the firn), we compared our lake dataset to modelled surface melt from RACMO 2.4p1. Multiple large (>4 mm w.e. day $^{-1}$) but generally short lived (≤1 week) spikes in melt are modelled, separated by periods of

little-to-no surface melt (Fig. 9). A correspondence between peak lake area, volume and surface melt is particularly apparent in 2014, 2015, 2017 and 2018 (Fig. 9). For example, in 2014-2015, total lake area (18.3 km$^2$) and volume (31.5 x10$^6$ m$^3$) peak five days after a large surface melt event (9 mm w.e day $^{-1}$; Fig. 9b). Similarly, in 2017-2018, total lake area (12.4 km$^2$) and volume (15.2 x10$^6$ m$^3$) peak eight days after a spike in surface melt (11 mm w.e day $^{-1}$; Fig. 9d). Though total lake area and volume may have peaked more rapidly following these modelled surface melt events, temporal gaps in cloud-free imagery

prevent us from constraining these timings further. Unfortunately, we have insufficient observations to detect whether a correspondence exists between peak lake area, volume and surface melt in other melt seasons (Supplementary Fig. 5). We also note that at the time of submission, RACMO2.4p1 data for the 2019-2020 melt season are unavailable which prevents us from comparing surface melt with total lake area and volume over this period. According to simulations by RACMO 2.4p1 over the whole ice shelf, total lake area is not significantly correlated to minimum firn air content (FAC; r$^2$ 0.01, p = 0.56) and mean

FAC (r$^2$ 0.009, p = 0.63) during the years for which we had lake observations (Fig. 10a-b). However, there is a qualitative correspondence between years of particularly low mean and minimum FAC and years of high melt (2014, 2015 and 2017; Fig. 10c). During these years, mean FAC ranged from 14.5 to 14.59 m and minimum FAC did not exceed 0.31 m.

**5 Discussion**

**5.1 Spatial distribution of SGLs**

The clustering of SGLs in the grounding zone of Shackleton Ice Shelf can be explained by localised albedo-lowering feedbacks. Extensive blue ice areas upstream of the grounding line and adjacent to the Bunger Hills nunataks (Fig. 1) are indicative of persistent snow surface scouring by easterly katabatic winds. These winds descend from the polar plateau, generated by large surface slopes (>20°) and reach 19 m s $^{-1}$ in summer (Doran et al., 1996). Surface meltwater production is enhanced as the winds warm adiabatically as they descend to the grounding zone, which lowers humidity and raises near-surface air

temperatures (Doran et al., 1996; Lenaerts et al., 2017). The lower albedo of blue ice areas (~0.5) compared to refrozen snow (~0.7) exerts a positive feedback on surface melting by enhancing solar radiation absorption (Kingslake et al., 2017; Lenaerts et al., 2017; Stokes et al., 2019). SGLs have been associated with strong katabatic wind-induced scouring of blue ice areas on several other East Antarctic ice shelves (Bell et al., 2017; Kingslake et al., 2017; Lenaerts et al., 2017; Stokes et al., 2019). This albedo-lowering feedback is locally enhanced by the presence of the Bunger Hills and surrounding nunataks (Fig. 1),

whose lower albedo further enhances surface melting and ponding on these portions of the ice shelf. This is a typically localised effect (i.e. within 10 km; Kingslake et al., 2017; Leppäranta et al., 2013; Winther et al., 1996; Stokes et al., 2019).





It is also clear that firn air content (FAC) is an important control on the spatial distribution of SGLs on Shackleton Ice Shelf. Lake clustering near the grounding line means repeated meltwater percolation and refreezing into the snow and firn layers is

likely to have formed refrozen ice lenses or superimposed ice zones (Hubbard et al., 2016). The large coalesced lakes we observed in 2014, 2015 and 2020 (e.g. Fig. 2d) are suggestive of the firn layer becoming saturated in this portion of the shelf. Simulations from RACMO 2.4p1 indicate that the FAC may be as low as ≤1 m on parts of the ice shelf (Fig. 10a, 10c). SGLs could therefore become more extensive in the ice shelf grounding zone in future, because saturated firn promotes surface ponding by limiting meltwater percolation and its lateral flow within the firn layer (Alley et al., 2018; Hubbard et al., 2016;

Lenaerts et al., 2017). The absence of SGLs further down-ice from the grounding zone can be explained by increased accumulation (snowfall) rates towards the ice shelf calving front which result in a corresponding higher FAC, meaning surface meltwater can be retained in a deeper snowpack and firn before it can pond into SGLs (Bell et al., 2017; Stokes et al., 2019). Thus, the clustering of SGLs around the grounding line is suggestive of air-depleted firn, which acts as a primary control on where meltwater can pond on the ice shelf surface.


At a more local scale, lakes on grounded ice upstream of the ice shelf grounding line have a high tendency to reform annually in the same locations (Fig. 2), which we attribute to the glaciological processes controlling SGL formation. Ice surface depressions on grounded ice are translations of subglacial topographic undulations, and are therefore fixed in space rather than migrating with ice flow (Echelmeyer et al., 1991; Lampkin and VanderBerg 2011; Langley et al., 2016). Conversely, lakes on

the ice shelf itself are generally shorter-lived because they form in surface depressions that migrate annually with ice flow and are progressively advected towards the ice shelf front, as on other Antarctic ice shelves (Banwell et al., 2014; Bell et al., 2017; Glasser and Gudmundsson 2012; Luckman et al., 2014; Reynolds and Smith 1981). We speculate that surface undulations perpendicular to the grounding line (i.e. parallel to ice flow) may form as ice flow converges when it crosses the break in slope and is deflected around the Bunger Hills (Fig. 2c), which could flex and fracture the ice (Banwell et al., 2014; Glasser and

Gudmundsson, 2012).

Though lakes on the ice shelf are commonly more transient and appear in different locations each year, some larger lakes on the eastern portion of the ice shelf and on Apfel Glacier have a high tendency to reform annually in the same locations (Fig. 2b and 2c). This characteristic has not previously been reported, and we suggest this occurs because these lakes are located on

very slow-moving ice near the grounding line (<50 m yr $^{-1}$) where annual ice advection is minimal. Between 2000-2020, SGLs on the ice shelf are only slightly shallower than those upstream of the grounding line, as has been recorded on Petermann Glacier in Greenland (Supplementary Fig. 4a-b; MacDonald et al., 2018). Furthermore, in years with higher meltwater volumes, lakes grew deeper on the ice shelf than on grounded ice (Supplementary Fig. 4c-d). This is perhaps surprising as slow ice flow produces low-amplitude ice surface topography and shallower depressions, so we would expect grounded lakes

to be deeper (Pope et al., 2016). We suggest greater numbers of deeper structural depressions form in the ice shelf grounding zone where there is an extensional flow regime and very low slopes conducive to meltwater ponding (Fig. 3b).



The spatial distribution of SGLs mapped in this study closely follows predicted meltwater routing pathways derived from a surface DEM, which assumes widespread firn saturation across the ice shelf (Supplementary Fig. 6). Hydrological routing
analysis shows meltwater coalesces from steeper upstream parts of the ice shelf and converges in structurally-determined topographic lows, such as flow stripes and crevasses on faster-flowing ice, downstream of ice rises, and in shear margins (Supplementary Fig. 6). This could have future implications for ice shelf stability (Sect. 5.3).

## 5.2 Seasonal evolution of SGLs

SGLs on Shackleton Ice Shelf exhibit strong intra-seasonal and inter-seasonal variations in area, depth and volume, as do
studies on Langhovde Glacier and the Nansen and Amery ice shelves (Bell et al., 2017; Langley et al., 2016; Moussavi et al., 2020). In melt seasons when lakes store larger total volumes of surface meltwater on Shackleton Ice Shelf, lakes are more extensive and some are fed by supraglacial rivers, which are not present in lower-melt years (Fig. 4, Fig. 6g-j). Notably, we find a correspondence between short-lived, high magnitude modelled snowmelt events and peak total lake area and volume during years of high surface meltwater storage (i.e. the melt seasons with maximum lake area, depth and volume: 2013-2014,
2014-2015, 2016-2017 and 2017-18) (Fig. 9). These spikes in modelled snowmelt could represent localised katabatic wind-induced melt events, similar to föhn wind-induced melt events on the Antarctic Peninsula modelled by regional climate models (Kuipers Munneke et al., 2018; Tuckett et al., 2019). Although 2019-2020 modelled surface melt data is currently unavailable for us to compare with total lake area and volume, we note that this was an exceptional melt season (Fettweis, 2020) in which we record the highest total lake area and volume over the 2000-2020 period (Fig. 9e). Previous work based on meteorological
observations from Ekström Ice Shelf, East Antarctica, has suggested that snowmelt-albedo feedbacks exert a key control on melt intensity and duration (Jakobs et al., 2019). Therefore, variations in the strength of this snowmelt-albedo feedback could exert an important control on the amount of melt seasonally available to pond into SGLs on Shackleton Ice Shelf.

Interestingly, the years when total lake area, depth and volume were highest do not always correspond with warmer summer
near-surface temperatures (Fig. 7). In particular, mean DJF temperatures during years with the highest surface meltwater storage were actually lower than some years with much less extensive and shallower lakes (Table 2). Furthermore, the weak positive relationship we observe between total SGL area and mean surface air temperature in the preceding part of the melt season (Fig. 8) contrasts with previous studies which show SGLs are highly sensitive to small fluctuations in surface temperature (e.g. Langley et al., 2016). Rather, we suggest that seasonal fluctuations in lake area and volume on this ice shelf
are more sensitive to snowmelt intensity associated with katabatic wind-driven melting, rather than longer-term patterns of mean near-surface summer temperature. Therefore, intense short-lived melt events appear to determine seasonal variability in SGL extent and volume. This is important because short-lived katabatic events and albedo feedbacks are difficult to resolve in regional climate models such as RACMO, owing to its 27-km resolution (Van Wessem et al., 2018). This finding underscores




the need for finer resolution regional climate model simulations of surface melt when considering future lake distributions
(Arthur et al., 2020; Stokes et al., 2019).

Regarding individual lake development, the lakes that we observed to drain vertically within a ≤ 7-day period did so more
rapidly than lakes on the Amery Ice Shelf where lakes have drained within 10 days (Moussavi et al., 2020). This is still slower
than on the Greenland Ice Sheet, where lakes drain on hourly-daily timescales (e.g. Das et al., 2008; Doyle et al., 2013; Selmes
et al., 2013; Stevens et al., 2015; Williamson et al., 2017). We note that the lake drainages we observed could have drained on
sub-daily timescales comparable to Greenland, but this is not detectable from the temporal resolution of the satellite imagery
used in this study. Ice shelf thicknesses in the region where lakes drained englacially (Fig. 6k-l) reach 150 - 200 m, which is
comparable to the ~200 m thickness of Larsen B through which lakes are thought to have drained prior to its collapse (Banwell
et al., 2013; MacAyeal and Sergienko 2013; Scambos et al., 2003). Therefore, lakes could potentially have drained through
the full ice shelf thickness via lake bed crevasses. However, because no lakes drained fully, we suggest they are more likely
to have drained into the firn and subsequently refrozen.

Most lakes do not drain and instead refreeze at the end of the melt season, which is consistent with the decline in total lake
area and volume from late-January to late-February (Fig. 7) and is common elsewhere in East Antarctica (e.g. Langley et al.,
2016; Leppäranta et al., 2013). Once meltwater penetrates the snowpack and reaches a sub-freezing firn temperature, it
refreezes and raises the surrounding firn temperature by releasing latent heat (Buzzard et al., 2018; Bevan et al., 2018; Jakobs
et al., 2019). Meltwater refreezing within the firn produces larger snow grains with a lower albedo, which are more likely to
absorb incoming solar radiation and further enhance surface melting (Gardner and Sharp 2010). Thus, our observations of
lakes repeatedly refreezing at the end of each melt season are likely to indicate meltwater storage and refreezing in the ice
shelf firn layer near the grounding line.

### 5.3 Implications for ice shelf stability

Future warming is expected to enhance surface melting and increase surface meltwater production in Antarctica (Bell et al.,
2018; Kingslake et al., 2017; Rintoul et al., 2018; Trusel et al., 2015). Increased lake coverage and meltwater storage in future
years could cause the ice shelf to flex (Banwell and MacAyeal 2015; Banwell et al., 2019), which could lead to fracture.
Previous work has suggested that FAC can used to infer shelf stability and likelihood of hydrofracture (Alley et al., 2018;
Holland et al., 2011; Lenaerts et al., 2017). We find simulated minimum FAC is close to zero in the grounding zone of
Shackleton Ice Shelf, consistent with simulated FAC on the Roi Baudouin (Lenaerts et al., 2017) and pre-collapse Larsen B
(Leeson et al., 200) ice shelves. Therefore, despite limited (<1 %) lake coverage on the ice shelf at present, its susceptibility
to widespread ponding and hydrofracture is likely to gradually increase as FAC is depleted (Alley et al., 2018; Leeson et al.,
2020; Lenaerts et al., 2017). Our meltwater routing analysis suggests that the ice shelf could potentially support and route





water along surface troughs to the ice shelf edge, similar to active shear-margin rivers present on Nansen Ice Shelf (Supplementary Fig. 6; Bell et al., 2017), which could partially mitigate the effect of surface meltwater loading (Bell et al., 2017).

## 6 Conclusions

This study presents the first quantitative, multiyear study of SGL evolution on East Antarctica's sixth largest ice shelf. Between the melt seasons of 1999-2000 and 2019-2020, melt seasons were characterised by hundreds of SGLs that were, on average, 0.02 km$^2$ in area, 0.96 m deep, and held a total meltwater volume of 7.45 x10$^6$ m$^3$. We also find that:

- Lakes cluster around the grounding line, controlled by localised albedo-lowering feedbacks caused by interactions between katabatic-driven melting, blue ice and exposed bedrock (Kingslake et al., 2017; Lenaerts et al., 2017; Stokes
et al., 2019).
- Lakes on the ice shelf are more transient than those on grounded ice, which reform annually in the same locations. Some lakes on slow-moving parts of the ice shelf near the grounding line also frequently occupy the same ice surface depressions, which likely results from minimal ice advection in these areas.
- Lakes begin to form from mid-November and their total volume peaks in early-late January each year, which reached
a maximum of 28.4 x10$^6$ m$^3$ on 31st January 2020.
- SGLs were most extensive, deepest, and formed at the highest recorded elevation (527 m) in 2014, 2015, 2017, 2018 and 2020. At the peak of these melt seasons, parts of the ice shelf grounding zone became saturated with meltwater which coalesced into large lakes, and lakes were fed by supraglacial rivers.
- Short-lived high magnitude snowmelt events correspond with peak total lake area and volume during these years of
high surface meltwater storage. Seasonal variability in lake extents is more sensitive to the intensity of individual melt events rather than mean near-surface summer temperatures, which we suggest is driven by localised katabatic wind-induced melting.

Our findings may be used to inform the boundary conditions and validation of supraglacial hydrology models, for example to
validate estimates of lake distributions and to constrain volume-dependent lake drainage thresholds. In particular, our data quantify the meltwater volumes being stored on the ice shelf surface, highlighting the importance of increased meltwater storage within SGLs during melt seasons experiencing intense, short-lived melt events. An improved understanding of East Antarctica's present surface hydrology has significance for constraining future dynamic change of the ice sheet and its contribution to sea-level rise.




**Figure 1. Overview map of Shackleton Ice Shelf with LIMA mosaic (Bindschadler et al. 2008) in the background. The grounding line (thick black line) is taken from Rignot *et al.* (2016). The subset area used for detailed analysis of lake area and depth extractions is outlined in green and the black dashed box outlines the area used to extract ERA5 reanalysis near-surface air temperature. The light blue areas visible on the ice sheet are blue-ice areas.**





**Figure 2. Recurrence frequency of lakes mapped between 1974 and 2020 on Shackleton Ice Shelf out of a total of 58 individual satellite scenes. Surface melt flux derived from QuickScat scatterometer (2000 - 2009 average from Trusel *et al.* (2013)) on the portion of the ice shelf where lakes predominantly form (a). Lakes are concentrated in the grounding zone downstream of blue ice and adjacent to exposed rock (b and c). Lakes on the ice shelf itself are short-lived in a given location (d) and migrate down-shelf with ice flow over multiple melt seasons (f). Lakes on grounded ice upstream of the grounding line typically occupy fixed ice surface depressions and therefore frequently reform in the same locations during multiple melt seasons (e). Note the scale bar frequency corresponds to number of times a lake was recorded in the available cloud-free satellite imagery, rather than the number of years over this time period.**





555

Figure 3. Frequency distributions of supraglacial lakes on Shackleton Ice Shelf by topographic variables: (a) individual lake mean elevations; (b) ice surface slope of individual lakes; (c) ice flow speed; (d) distance of each lake to nearest exposed bedrock. The four panels show a subset of lakes (in blue) on the ice shelf and their spatial association with low elevations, low surface slopes low ice flow speeds and exposed rock/ blue ice. The grounding line is represented by the dashed black line.

560









600 **Figure 4. Evolution of total lake area, depth and volume across Shackleton Ice Shelf over the 2000-2020 melt seasons. The stem plots present time series of meltwater area (a), depth (b) and volumes (c) of lakes inside the subset area (green box in Figure 1), where each point and stem represent a single date of lake observations. Grey bars indicate the percentage of cloud cover within this subset area.**

605

610


615

**Figure 5. Total lake volume anomaly relative to long-term (2000-2020) mean (a) and total lake area anomaly relative to long-term (2000-2020) mean (b). Dashed black lines represent one, two and three standard deviations from the long-term mean.**









**Figure 6. Examples of lake drainage and refreeze-burial events on Shackleton Ice Shelf: lakes refreezing and becoming buried by snow (a-c); a lake draining via a supraglacial stream (d-f); supraglacial streams/rivers feeding lakes (g); a lake before and after through-ice drainage (h-j); and a chain of lakes partially draining in ≤ 7 days (k-l). Background images are Landsat 8 (a-c) and Sentinel 2A (d-l).**













**Figure 7. Total supraglacial lake area and mean near-surface (2 m) temperature for 1999 – 2004 (a), 2010 – 2015 (b) and 2015 – 2020 (c). Mean near-surface temperature is the mean extracted over the whole Shackleton Ice Shelf (shown as a 5-day moving average).**




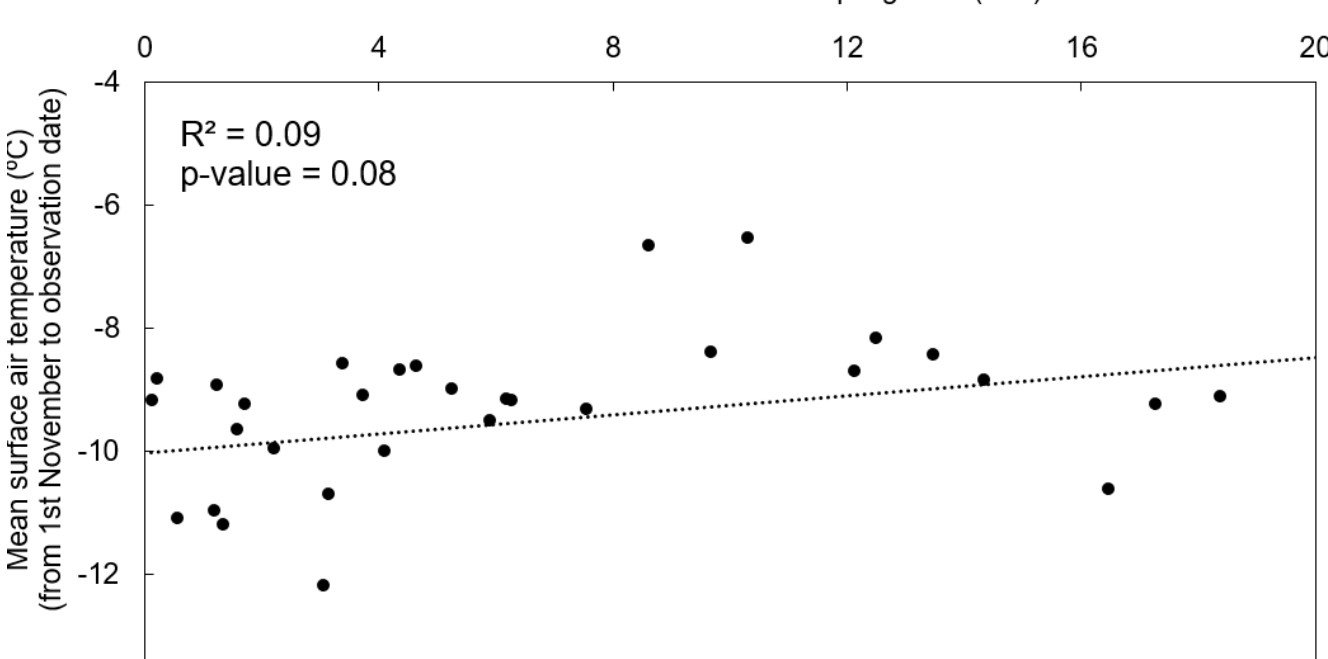

**Figure 8. Scatter plot of total lake area inside the subset area highlighted in Figure 1 (green box) and mean surface air temperature from 1st November to the date of each observation.**










**Figure 9. Seasonal variations in total lake area, volume and modelled snowmelt on Shackleton Ice Shelf during the melt seasons 2013/14 (a), 2014/15 (b), 2016/17 (c), 2017/18 (d) and 2019/20 (e) within the subset area (green box, Figure 1). Peak total lake area and volume is associated with a spike in snowmelt. Snowmelt rates are mean values over the entire ice shelf.**

**Figure 10. (a) Mean firn air content (total integrated pore space, i.e. the thickness of the equivalent air column contained in the firn in metres) over Shackleton Ice Shelf from 2000-2019. (b) Scatterplot comparing minimum fir air content and total lake area. (c) Time series comparing mean firn air content, minimum firn air content and total lake area.**



**Table 1. Total lake area and volume within the 48 x 48 km² subset area on Shackleton Ice Shelf.**

| | Lake depth (m) | | | Lake area (km²) | | | | Lake volume (x10⁶ m³) |
|---|---|---|---|---|---|---|---|---|
| Date | Median | Mean | Maximum | Mean | Median | Maximum | Total | Total |
| 6ᵗʰ Feb 2000 | 0.69 | 0.69 | 1.52 | 0.014 | 0.004 | 0.203 | 1.19 | 0.85 |
| 8ᵗʰ Feb 2004 | 0.5 | 0.55 | 2.17 | 0.028 | 0.004 | 0.986 | 13.47 | 8.64 |
| 27ᵗʰ Jan 2008 | 0.78 | 0.78 | 3.27 | 0.022 | 0.008 | 0.459 | 4.09 | 3.48 |
| 17ᵗʰ Feb 2010 | 0.59 | 0.68 | 2.51 | 0.028 | 0.007 | 0.364 | 3.71 | 2.34 |
| 24ᵗʰ Feb 2010 | 0.48 | 0.49 | 0.95 | 0.026 | 0.007 | 0.376 | 5.88 | 3.20 |
| 4ᵗʰ Feb 2011 | 0.71 | 0.77 | 2.47 | 0.031 | 0.009 | 0.359 | 6.26 | 4.95 |
| 23ʳᵈ Feb 2012 | 0.57 | 0.57 | 2.21 | 0.015 | 0.004 | 0.139 | 1.33 | 0.72 |
| 2ⁿᵈ Dec 2013 | 0.96 | 0.98 | 3.58 | 0.004 | 0.001 | 0.178 | 3.04 | 2.82 |
| 3ʳᵈ Jan 2014 | 1.80 | 1.78 | 6.56 | 0.026 | 0.002 | 1.867 | 16.45 | 33.94 |
| 19ᵗʰ Nov 2014 | 0.70 | 0.69 | 1.00 | 0.031 | 0.007 | 0.171 | 0.56 | 0.41 |
| 6ᵗʰ Jan 2015 | 1.72 | 1.78 | 6.46 | 0.045 | 0.005 | 6.717 | 18.37 | 31.50 |
| 7ᵗʰ Feb 2015 | 1.57 | 1.51 | 3.13 | 0.017 | 0.002 | 0.949 | 3.38 | 4.67 |
| 23ʳᵈ Feb 2015 | 0.89 | 0.98 | 2.09 | 0.044 | 0.016 | 0.196 | 1.19 | 1.07 |
| 9ᵗʰ Jan 2016 | 1.77 | 1.87 | 5.72 | 0.006 | 0.001 | 0.182 | 1.57 | 2.93 |
| 25ᵗʰ Jan 2016 | 1.84 | 1.91 | 5.61 | 0.009 | 0.003 | 0.223 | 2.19 | 4.38 |
| 26ᵗʰ Feb 2016 | 1.16 | 1.25 | 3.74 | 0.012 | 0.002 | 0.277 | 3.13 | 3.62 |
| 27ᵗʰ Jan 2017 | 1.35 | 1.44 | 3.99 | 0.019 | 0.002 | 1.464 | 10.29 | 16.35 |
| 31ˢᵗ Jan 2017 | 0.63 | 0.71 | 3.09 | 0.020 | 0.002 | 1.573 | 8.60 | 7.19 |
| 28ᵗʰ Feb 2017 | 0.46 | 0.58 | 2.23 | 0.041 | 0.003 | 0.477 | 1.71 | 0.89 |
| 29ᵗʰ Dec 2017 | 1.52 | 1.66 | 3.41 | 0.002 | 0.001 | 0.022 | 0.16 | 0.26 |
| 19ᵗʰ Jan 2018 | 0.97 | 0.61 | 1.05 | 0.007 | 0.001 | 0.684 | 12.48 | 15.25 |
| 26ᵗʰ Jan 2018 | 0.86 | 0.90 | 3.89 | 0.009 | 0.001 | 0.729 | 9.66 | 9.41 |
| 8ᵗʰ Feb 2018 | 0.46 | 0.52 | 3.02 | 0.008 | 0.001 | 0.876 | 4.34 | 2.26 |
| 15ᵗʰ Feb 2018 | 1.03 | 1.29 | 4.47 | 0.017 | 0.003 | 0.081 | 0.20 | 0.43 |
| 29ᵗʰ Jan 2019 | 1.17 | 1.21 | 2.69 | 0.019 | 0.003 | 0.656 | 5.24 | 6.36 |
| 18ᵗʰ Feb 2019 | 0.53 | 0.52 | 1.58 | 0.031 | 0.001 | 1.50 | 7.54 | 4.48 |
| 28ᵗʰ Feb 2019 | 0.52 | 0.53 | 1.87 | 0.047 | 0.008 | 1.494 | 6.18 | 3.45 |



| | | | | | | | | |
|---|---|---|---|---|---|---|---|---|
| **19ᵗʰ Dec 2019** | 0.59 | 0.88 | 2.15 | 0.023 | 0.011 | 0.522 | 4.63 | 2.89 |
| **1ˢᵗ Jan 2020** | 0.91 | 0.95 | 5.23 | 0.012 | 0.001 | 0.954 | 12.12 | 14.22 |
| **11ᵗʰ Jan 2020** | 0.71 | 0.73 | 3.47 | 0.019 | 0.001 | 1.627 | 17.25 | 16.48 |
| **31ˢᵗ Jan 2020** | 0.69 | 0.76 | 2.94 | 0.045 | 0.001 | 11.961 | 28.55 | 28.44 |
| **28ᵗʰ Feb 2020** | 0.41 | 0.41 | 0.60 | 0.082 | 0.008 | 3.204 | 14.32 | 0.67 |


**Table 2. Average December-January-February (DJF) near-surface temperatures from ERA5 on Shackleton Ice Shelf from 2000-2020.**

| Melt season | DJF Temperature (˚C) | Melt season | DJF Temperature (˚C) |
|---|---|---|---|
| 1999-2000 | -10.0 | 2009-2010 | -8.9 |
| 2000-2001 | -7.9 | 2010-2011 | -9.4 |
| 2001-2002 | -7.5 | 2011-2012 | -10.3 |
| 2002-2003 | -8.1 | 2012-2013 | -9.6 |
| 2003-2004 | -7.2 | 2013-2014 | -8.6 |
| 2004-2005 | -7.1 | 2014-2015 | -8.4 |
| 2005-2006 | -8.1 | 2015-2016 | -10.2 |
| 2006-2007 | -7.9 | 2016-2017 | -6.4 |
| 2007-2008 | -10.6 | 2017-2018 | -8.6 |
| 2008-2009 | -8.0 | 2018-2019 | -8.3 |
| | | 2019-2020 | -9.3 |





**Data Availability**

The mapped supraglacial lake polygons are available as digital GIS shapefiles (.shp) in Supplementary Data. Sentinel imagery
is available from the Copernicus Open Access Hub (https://scihub.copernicus.eu) and Landsat scenes are available from the
United States Geological Survey EarthExplorer (https://earthexplorer.usgs.gov/). ERA5 reanalysis surface temperature is
available from the Copernicus Climate Change Service Climate Date Store (https://cds.climate.copernicus.eu/). We
acknowledge the Norwegian Polar Institute's Quantarctica package.

**Author contribution**

JFA, CRS and SSJ designed the initial study. JFA undertook the data collection and conducted the analysis, with guidance
from all authors. JFA led the manuscript writing, with input from all authors.

**Competing Interests**

The authors declare that they have no conflict of interest.

**Acknowledgements**

JFA was funded by the IAPETUS Natural Environment Research Council (NERC) Doctoral Training Partnership (grant
number NE/L002590/1). CRS and SSRJ were supported by NERC grant NE/R000824/1. We acknowledge RACMO data from
M.R van den Broeke, the Norwegian Polar Institute's Quantarctica package, and Copernicus Climate Change Service
Information (ERA5 reanalysis). JFA acknowledges BWJ Miles for helpful discussions during this work.

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
