# Peer review of "Distribution and seasonal evolution of supraglacial lakes on Shackleton Ice Shelf, East Antarctica"

_The Cryosphere, 2020_

## Referee Comment (RC1) · Jan Lenaerts (Referee) · 4 Jun 2020

Arthur et al. present a data set of long-term (2000-2020) remote sensing observations of lake extent, depth and volume on Shackleton ice shelf East Antarctica. They find that lakes predominantly form and exist in the ice shelf grounding zone, and that lake extent displays substantial seasonal and interannual variations. They do not find a clear connection between summer temperature or melt and lake extent, which suggests that lake formation is likely associated to episodic strong melt events. This is a well-written paper, and the methods and results (including uncertainties and limitations) are presented in a clear and coherent way. I think it is suitable for publication in

[Figure]

The Cryosphere after some revisions, mainly clarifications and some extension of the discussion. I have highlighted some ideas below.

L67: the ice shelf itself. . . Perhaps rewrite to: 'the ice shelf surface flow has accelerated' or 'speed has increased'. . .

L170: we found a mean absolute error of 0.007 and a root mean square error of 0.029 between manually-digitised SGL areas and those derived from NDWI What are the units?

L178: just out of curiosity, would you expect variability in lake bottom albedo based on its proximity to rock outcrops, due to eolian deposition of dark material on the surface that is then collected in the lake?

L206: 0.25 degree in a regular grid translates to a much higher longitudinal resolution at high latitudes.

L213: The routing analysis is useful, but is highly simplified since it does not allow for flow in the firn and subsequent storage. This should be clearly indicated here, and/or in the discussion.

L251: likely because of its northerly location, allowing an earlier start of the melt season?

L275: in principle, lakes could also drain vertically via fractures. This is discussed later, but not mentioned here.

L407: The authors briefly discuss the poor correlation between lake extent and temperature and melt. I would agree that these results are not surprising, given the fact that ERA5 nor RACMO2 (at this resolution) cannot resolve the atmospheric phenomena around the grounding zone (katabatic winds, enhanced turbulent mixing around slope break from grounded to floating ice), and -at least as important- do not account for the presence of blue ice or outcrops and their low surface albedo. So, while lake formation might be more associated to episodic melt events – as the authors suggest, since the

datasets used here do not even represent the climatology and spatial heterogeneity correctly, it is hard to proof either way. Perhaps it's worthwhile to extend this discussion somewhat.

---

## Referee Comment (RC2) · Anonymous Referee #2 · 4 Jun 2020

Using optical satellite imagery, analysed alongside modeled temperature and surface melt data, this paper presents a detailed study of surface lake evolution over the last two decades on the Shackelton Ice Shelf, East Antarctica. The authors observe extensive ponding in the region, and from their analysis find that katabatic winds and an albedo feedback play a key role in the formation of lakes, and the timing of variations in extent and volume.

The method used to analyse changes in lake characteristics is scientifically sound, and the analysis of this data alongside climatic factors is of a high quality and interesting. The paper is also clear in stating its limitations. This detailed and comprehensive anla-

ysis of lake evolution, both through and between seasons, at a particular region, will be valuable to the scientific community as we strive to improve our understanding of surface hydrology in the Antarctic and elsewhere. A detailed temporal study of this nature is particularly welcome. The results and discussion will especially be of value to those studying the nature of lake evolution and the factors associated with that evolution, including modelers.

I do think the clarity of the paper in sections needs to be improved before final publication. In particular, I think there are instances where lack of consistency and specificity of language can hinder clear understanding. I attach an annotated PDF with all my comments and suggestions. Briefly, some key issues and suggestions are:

-Section 3.3 needs substantial change in terms of structure and the use of terms. See the PDF.

-I think the references to 'englacial' drainage, and what is meant by that, need to be made clearer. See the PDF.

- The information in the paragraph from L45 to L56 seems as though it should be in the study site section. Currently 'study site' seems spread out over two sections in a way that disrupts the flow.

- It would be interesting if the authors, in their discussion, assessed the modeled findings of Banwell and MacAyeal (2015) about lakes deepening inter-annually on ice shelves. I only suggest that they briefly assess Banwell/MacAyeal's finding against this paper's analysis.

-I think figure 4 needs work to be more useful to readers. I find that the attempt to include data from every year in the long study period makes it almost impossible to tell which color corresponds to which year. I leave it to the authors to decide how best to overcome this – perhaps they need to be selective about which data from this series is most important to demonstrate the findings and move some to a supplementary figure.

I appreciate this is a difficult challenge. I also have a small suggestion for figure 1 and a tiny cosmetic suggestion for figure 10. All suggestions are included in the same pdf.

Thank you very much to the authors for an interesting read and for contributing this valuable research.

Please also note the supplement to this comment:
https://www.the-cryosphere-discuss.net/tc-2020-101/tc-2020-101-RC2-supplement.pdf

**Supplement:**

[revised manuscript text omitted]

---

## Author Comment (AC1) · 26 Jun 2020

Arthur et al. present a data set of long-term (2000-2020) remote sensing observations of lake extent, depth and volume on Shackleton ice shelf East Antarctica. They find that lakes predominantly form and exist in the ice shelf grounding zone, and that lake extent displays substantial seasonal and interannual variations. They do not find a clear connection between summer temperature or melt and lake extent, which suggests that lake formation is likely associated to episodic strong melt events. This is a well-written paper, and the methods and results (including uncertainties and limitations) are presented in a clear and coherent way. I think it is suitable for publication in The Cryosphere after some revisions, mainly clarifications and some extension of the discussion. I have highlighted some ideas below.

*We would like to thank the Reviewer for their work on our manuscript and their encouraging comments.*

L67: the ice shelf itself. . . Perhaps rewrite to: 'the ice shelf surface flow has accelerated' or 'speed has increased'. . .

*Amended: changed to 'the ice shelf surface flow has accelerated'.*

L170: we found a mean absolute error of 0.007 and a root mean square error of 0.029 between manually-digitised SGL areas and those derived from NDWI What are the units?

*Amended: units added (square kilometres, km$^2$).*

L178: just out of curiosity, would you expect variability in lake bottom albedo based on its proximity to rock outcrops, due to eolian deposition of dark material on the surface that is then collected in the lake?

*Agreed, enhanced lake bottom albedo and surface ablation have been recorded in western Greenland (Tedesco et al., 2012) and is acknowledged to be an important process on McMurdo Ice Shelf (MacAyeal et al., 2019; Glasser et al., 2014, Banwell et al., 2017; 2019; Macdonald et al., 2020), which is heavily covered with fine debris due to sediment redistribution from medial moraine and rock outcrops. Whilst Shackleton Ice Shelf is mostly 'clean' blue ice or snow-covered, we do observe evidence of wind-blown material in meltwater adjacent to the Bunger Hills rock outcrop in our study area visible in Google Earth imagery, though this tends to be in very small ponds rather than the larger supraglacial lakes. We have added some additional sentences to reflect this in the second paragraph of Section 3.4.*

L206: 0.25 degree in a regular grid translates to a much higher longitudinal resolution at high latitudes.

*We agree that this is the case, and over Antarctica this equates to ~31 km (Tetzner et al., 2019).*

L213: The routing analysis is useful, but is highly simplified since it does not allow for flow in the firn and subsequent storage. This should be clearly indicated here, and/or in the discussion.

*Amended, sentence updated at Line 382: 'This could have future implications for ice shelf stability, though we note that our routing analysis does not allow for water storage in the firn (Sect. 5.3).'*

L251: likely because of its northerly location, allowing an earlier start of the melt season?

*We agree that this is likely to be the case, and have added a sentence at Line 253: 'We suggest the earlier onset of lake formation reflects an earlier start to the melt season due to the northerly location of this ice shelf.'*

L275: in principle, lakes could also drain vertically via fractures. This is discussed later, but not mentioned here.

*We have amended this sentence to 'We observe lakes disappearing in three ways', to clarify that these are the three mechanisms we observe in this study, while discussing vertical lake drainage in Section 5.2.*

L407: The authors briefly discuss the poor correlation between lake extent and temperature and melt. I would agree that these results are not surprising, given the fact that ERA5 nor RACMO2 (at this resolution) cannot resolve the atmospheric phenomena around the grounding zone (katabatic winds, enhanced

turbulent mixing around slope break from grounded to floating ice), and -at least as important- do not account for the presence of blue ice or outcrops and their low surface albedo. So, while lake formation might be more associated to episodic melt events – as the authors suggest, since the datasets used here do not even represent the climatology and spatial heterogeneity correctly, it is hard to proof either way. Perhaps it's worthwhile to extend this discussion somewhat.

*This is a good point, which we have carefully considered. We used ERA5 in the absence of any long-term local meteorological observations within our 2000-2020 study period. ERA5 has been shown to be accurate in reproducing local climate variability on the Antarctic Peninsula, with only a small negative bias in near-surface air temperature (Tetzner et al., 2019). However, we acknowledge that the resolution of ERA5 cannot resolve these localised processes. Unfortunately, RACMO2 output is not yet available at the higher 5.5 km resolution over this region of East Antarctica, which could better resolve these localised processes. Nevertheless, to our knowledge this is the first time that seasonal supraglacial lake distributions are compared with regional climate model surface melt and (near)surface air temperature in East Antarctica. We think that the strong correspondence between peaks in supraglacial lake extents and peaks in RACMO surface melt is an important finding and indicates that episodic intense melt events are key on this ice shelf for lake formation and evolution. We have extended our discussion surrounding the correlation between lake extent, temperature and melt in Section 5.2.*

---

## Author Comment (AC2) · 26 Jun 2020

Reviewer 2 Comments:

Using optical satellite imagery, analysed alongside modeled temperature and surface melt data, this paper presents a detailed study of surface lake evolution over the last two decades on the Shackleton Ice Shelf, East Antarctica. The authors observe extensive ponding in the region, and from their analysis find that katabatic winds and an albedo feedback play a key role in the formation of lakes, and the timing of variations in extent and volume. The method used to analyse changes in lake characteristics is scientifically sound, and the analysis of this data alongside climatic factors is of a high quality and interesting. The paper is also clear in stating its limitations. This detailed and comprehensive analysis of lake evolution, both through and between seasons, at a particular region, will be valuable to the scientific community as we strive to improve our understanding of surface hydrology in the Antarctic and elsewhere. A detailed temporal study of this nature is particularly welcome. The results and discussion will especially be of value to those studying the nature of lake evolution and the factors associated with that evolution, including modelers.

I do think the clarity of the paper in sections needs to be improved before final publication. In particular, I think there are instances where lack of consistency and specificity of language can hinder clear understanding. I attach an annotated PDF with all my comments and suggestions.

*We would like to thank the Reviewer for their work on our manuscript and their constructive comments. We respond to their specific concerns below.*

-Section 3.3 needs substantial change in terms of structure and the use of terms. See the PDF.

*We have addressed the specific comments relating to this Section below.*

-I think the references to 'englacial' drainage, and what is meant by that, need to be made clearer. See the PDF.

*We have addressed this in Lines 275, 289 and 412, please see specific responses to comments below.*

- The information in the paragraph from L45 to L56 seems as though it should be in the study site section. Currently 'study site' seems spread out over two sections in a way that disrupts the flow.

*We have included this information in this paragraph as a justification for our selection of this particular ice shelf, and in particular to highlight its vulnerability to hydrofracturing, which is of key importance in the context of surface hydrology and ice shelf stability. Th purpose of this paragraph is also to introduce the ice shelf to the reader who may not be familiar with our study area. Our reasoning for including a separate Study Site section is to provide more specific glaciological context.*

- It would be interesting if the authors, in their discussion, assessed the modeled findings of Banwell and MacAyeal (2015) about lakes deepening inter-annually on ice shelves. I only suggest that they briefly assess Banwell/MacAyeal's finding against this paper's analysis.

*We have addressed this in Line 373, please see specific responses to comments below.*

-I think figure 4 needs work to be more useful to readers. I find that the attempt to include data from every year in the long study period makes it almost impossible to tell which color corresponds to which year. I leave it to the authors to decide how best to overcome this – perhaps they need to be selective about which data from this series is most important to demonstrate the findings and move some to a supplementary figure.I appreciate this is a difficult challenge. I also have a small suggestion for figure 1 and a tiny cosmetic suggestion for figure 10. All suggestions are included in the same pdf. Thank you very much to the authors for an interesting read and for contributing this valuable research.

*We considered these suggestions regarding Figure 4, but decided not to modify the data shown in this Figure. This is because we feel it is important to show the long-term (2000-2020) time series of total lake area, depth and volume, and so feel it would be inappropriate to selectively remove certain years into another Figure in the Supplementary Information. We feel the particular strength of this Figure is that it summarises both the long term and the seasonal evolution of supraglacial lakes. By plotting these data on a common seasonal timescale, it is easier to see the relationship between lake area, depth and volume.*

*Although individual years could potentially be made more distinguishable by using discrete contrasting colours, we think a graduated colour scale highlights data in later years, and in particular anomalously high years of lake meltwater storage (e.g. 2014, 2015, 2020). Full lake area, depth and volume data for individual years is contained in Table 1, and we have highlighted this in the figure caption. We have made the small suggestions to Figures 1 and 10, and thank the Reviewer again for their encouraging comments.*

Responses to specific comments (on PDF):

-Because 'melt' and 'pond' can both be used as nouns and verbs I think you have to be careful with their usage. In particular I think the first sentence needs to be revised, I had to read it several times and I am not sure it is necessary at all.

*L26: We agree the use of 'pond' as a verb in this first sentence is potentially confusing, and have decided to remove the sentence.*

This sentence sets up the expectation that you will outline the \*direct\* and \*indirect\* influences, but then you do not do that. I suggest explicitly doing that, now you don't mention an indirect one, or else you set it up differently.

*L27: We have added two sentences to explain how SGLs influences ice shelves both directly and indirectly – this relates to the comment on L34, please see below.*

-This sentence is one example of what was presented in the previous sentence – I don't think 'specifically' is appropriate.

*L28: Amended: deleted 'Specifically'.*

I suggest mentioning in addition to flexure, that lakes can also influence ice shelf stability by being a source of water to fill/propagate crevasses (e.g. Scambos et al., 2003). You do mention hydrofracture, but in passing elaboration of the flexure mechanism. I also suggest mentioning they influence the albedo of the surface.

*L34: We have added the following two sentences: 'SGLs can indirectly influence ice shelf dynamics by lowering surface albedo, which can intensify surface melt and induce a warming effect on the adjacent ice column (Lüthje et al., 2006; Tedesco et al., 2012; Hubbard et al., 2016). SGLs can also act as reservoirs by storing meltwater for crevasse penetration and hydrofracture (Scambos et al., 2000; 2003; 2009).'*

I suggest 'importance for'.

*L43: Amended: changed to 'importance for SGL evolution'.*

I find the placement of this paragraph a bit odd and disruptive to the flow given that the study site section comes up shortly after. I suggest moving this info into the study site section and only very briefly introducing the Shackleton in the following paragraph.

*L45: We have included this information in this paragraph as a justification for our selection of this particular ice shelf, and in particular to highlight its vulnerability to hydrofracturing which is of key importance in the context of surface hydrology and ice shelf stability. Our reasoning for including a separate Study Site section is to provide more specific glaciological context.*

I think over/above/below/under is more appropriate when discussing a threshold.

*L83: Amended: changed 'within' to 'below'.*

Would it not be advantageous and little work to use both Sentinel and Landsat in this period? i.e. use L8 when there is no S2?

*L89: We agree that it is advantageous to use a combination of Sentinel-2 and Landsat 8 imagery, and confirm that we used Landsat-8 where no suitable Sentinel-2 was available. We have clarified this in the text, which now reads: 'Sentinel-2A/B imagery was used in preference to Landsat 8 or another sensor, such*

*as the Moderate Resolution Imaging Spectroradiometer (MODIS) or Advanced Spacebourne Thermal Emission and Reflectance Radiometer (ASTER), owing to its high spatial resolution (10 m) and revisit period of 5 days, though Landsat 8 imagery was used where no Sentinel-2 imagery was available with suitable (≤ 20%) cloud cover.'*

It is not important, but the convention is to hyphenate Sentinel-2.

*L91: We have now hyphenated 'Sentinel-2' throughout the manuscript.*

I think it would be helpful for many readers to very briefly explain what a 'mixed pixel' is.

*L116: Amended to include a definition of 'mixed pixels', now reads: '(i.e. pixels containing a combination of water, slush and/or snow or ice […]'.*

Was shadowing an issue on any images?

*L119: Shadowing was largely not an issue in our lake classification. In some cases, small shadows associated with crevasses or individual rock outcrops were classified as water, and we removed these false positives following NDWI classification during final post-processing of the lakes dataset.*

Shouldn't the title be volumes and depths, not area?

*L127: Amended Section Heading to 'Extracting lake depths and volumes'.*

I assume you converted to TOA before carrying out the NDWI calculation – the flow of the text seems to suggest otherwise. You should mention this before the NDWI.

*L133: Agreed, we have moved this paragraph to Section 3.1 to improve the logical flow of the text.*

It is odd to mention that Rz uses the red band, but not Ad. Also, later you mention that you also use the pan band. I think this is quite confusing. I suggest not mentioning which band yet, just explaining the method.

*L140: Amended: removed 'red band', so sentence now reads: 'Rz is the reflectance value of a water-coloured pixel […]'.*

Banwell et al. (2019) takes a one-pixel ring around the water feature. Two pixels is fine, it just doesn't need to reference Banwell. In contrast, Sneed and Hamilton (2007) used the same Ad for the whole region, which could be mentioned. Again, don't mention the red band here – as its choice has not yet been explained, and also you use the pan band too.

*L149: Removed Banwell et al. (2019) citation and deleted 'red band'. We have also added the following sentence: 'This is an improvement on previous approaches which used static Ad values across a region (e.g. Sneed and Hamilton, 2007).'*

I find this paragraph to flit between topics a bit. Ad -> Band choice -> Rinf, then back to band choice in the following paragraph. Remove mention of band choice here and make it its own paragraph.

*L153: We have edited the structure of this paragraph by moving the sentence 'Red light attenuates more strongly in water than blue light, meaning that there are larger measurable changes in red reflectance over water than for other wavelengths (Box and Ski 2007; Pope et al., 2016)' to the beginning of the final paragraph in this section, to improve the logical flow from discussion of $A_d$, to $R_{inf}$, to choice of bands.*

Switch 'exposed to the atmosphere' to 'exposed at the surface'

*L154: Amended to: 'exposed at the surface'*

This is not the case in Banwell et al. (2014).

*L157: Removed Banwell et al. (2014) citation.*

Be consistent between Landsat and Sentinel in how you refer to bands – as colour, or number, or both. I suggest as in the following sentence – coluor with band number in brackets in the first instance, then just colour.

*L159: Amended to: 'red (Band 4)'.*

Again – what is colour is Landsat 7 band 3?

*L162: Amended to: 'red (Band 3) reflectances'.*

I think it is good to say above/below to be explicit about which side of the grounding line.

*L241: Amended 'beyond' to 'above'.*

Be more specific, offer a range?

*L249: Added: '(~100-1000)' so that sentence now reads as below.*

I think this 7.45x10^6 value is for mean total (at any one time) meltwater volume? I don't think that it's clear enough that 'on average' is coupled with this part of the sentence.

*L250: Amended: added 'an average', so that sentence now reads: 'In a typical melt season, we observed hundreds (~100-1000) of SGLs that were, on average, 0.02 km$^2$ in area, 0.96 m deep, and held an average total meltwater volume of 7.45 x10$^6$ m$^3$.'*

You later refer to this as englacial drainage, use the word englacial here.

*L275: Amended: added 'englacial' to 'drainage into the firn'.*

From optical imagery you can only be sure that the surface freezes over. Mention that it's the surface that freezes.

*L276: Amended to 'surface refreezing and/or snow burial'.*

I find the sentence from 'as indicated..' unclear, please rephrase.

*L277: Sentence re-worded to: 'Lakes commonly refreeze at the end of a melt season, indicated by relict frozen lake scars which are similar to refrozen lakes in previous studies'.*

Do you mean that the firn becomes saturated? 'portions of the ice shelf become saturated and pond into large lakes' – I think this is unclear.

*L282: Reworded to: 'During the peak of the melt season in some years, large portions of the ice shelf firn layer become saturated, causing meltwater to accumulate and coalesce in large lakes on portions of the ice shelf, such as on the tongue of Scott Glacier.'*

Explain exactly what you mean by englacially and how you can tell.

*L289: Added 'into the firn', sentence now reads: 'We also recorded several instances of lakes apparently draining englacially into the firn over a ≤ 7-day period during January 2018 and 2020'. We explain this interpretation in the subsequent two sentences.*

I think you are saying that if the chain of lakes had lost area due to refreezing rather than drainage, the surrounding lakes would also have lost area due to refreezing. However, I don't think this is clearly stated. Also be careful with the subject 'they' here and which set of lakes it is referring to.

*L291: Amended to: 'If this chain of lakes had refrozen, we would have expected to record a reduction in area of the surrounding lakes. However, the surrounding lakes did not change in area over this period, suggesting they also did not refreeze.'*

'Therefore, we interpret these lakes to have drained englacially into the firn'.

*L293: Sentence added: 'Therefore, we interpret these lakes to have drained englacially into the firn'.*

Katabatic winds are fundamental here so I suggest that you mention them in this first index sentence of the paragraph.

*L330: Amended: changed 'albedo-lowering' to 'katabatic wind-driven'.*

Could this also be related to inter-annual deepening as per Banwell and MacAyeal (2015)? I suggest that you assess the modelled finding in that paper about inter-annual deepening on floating ice against your results.

*L373: We agree that this could be related to inter-annual lake deepening, and have added the following sentence: 'Inter-annual lake deepening amplified by lake bottom ablation could also mean lakes on the ice shelf evolve to be deeper than those on ice upstream of the grounding line (Banwell and MacAyeal, 2015). We do not observe any inter-annual deepening of individual lakes in our study.'*

By lakes that drain vertically I think you are referring to the same group of lakes you earlier referred to as draining englacially – I don't think that is clear, please be careful and consistent with terms such as these. Otherwise it can seem as if you are introducing a different set/mechanism.

*L412: Amended: changed 'vertically' to 'englacially'.*

See above about this term/sentence construction. Part of the issue is the phrase 'that were, on average held a total meltwater volume' would not be grammatically correct. Unless I've misunderstood what that number represents, that phrase should work for the overall sentence to work properly.

*L447: Added 'mean', so that sentence now reads: 'Between the melt seasons of 1999-2000 and 2019-2020, melt seasons were characterised by hundreds of SGLs that were, on average, 0.02 km$^2$ in area, 0.96 m deep, and held a mean total meltwater volume of 7.45 x10$^6$ m$^3$.'*

I think the figure makes it appear as if the Denman Glacier is the boundary of the Shackleton and the area E of it is not part of the ice shelf, I suggest clarifying this in the figure. I would also appreciate a N arrow.

*L505: We agree that the label for Shackleton Ice Shelf could be construed as referring only to the part of the ice shelf west of Denman Glacier, and have added a second label to avoid confusion, as well as a north arrow.*

I suggest total 'lake' area to be consistent with 'lake depth'.

I appreciate that presenting 20 years of data is a challenge. Unfortunately, I think that it is virtually impossible to distinguish between some of these colours, and therefore I cannot interpret much of this graph. I will leave it to the authors to determine the best solution, but I do not think it works as it is. Consider picking out some of the most useful demonstrative data for a main fig and putting some in a supplementary figure.

*L570: We have amended the y axis title on Panel A as suggested, and amended the y axis title on Panel C to 'Total meltwater volume'. Regarding the Reviewer's suggestion for data display on this Figure, we have responded in the general comments above.*

In other instances the secondary y axis text is flipped the other way – it would be nice to flip this to be consistent.

*L730: Amended: rotated secondary y-axis label.*

---

## Author Response (AR1)

Jennifer Arthur
Department of Geography
South Road
Durham Dh1 3LE
Email: jennifer.arthur@durham.ac.uk

20th August 2020

Dear Stef Lhermitte,

RE: Response to Reviewer Comments

Thank you for your decision to accept our manuscript subject to minor revisions. Please find attached a revised version of the manuscript with tracked changes in red, in which we have implemented our responses to reviewers.

Thank you for taking your time to consider our revised manuscript.

Kind Regards,

Jennifer Arthur and co-authors.

[revised manuscript text omitted]